# ROBU-MARC: A MASKED AUTOENCODER-AIDED CAMERA-RADAR NETWORK FOR ROBUST 3D PERCEPTION UNDER SENSOR CORRUPTION

## ABSTRACT

Autonomous vehicles rely on robust perception systems, yet real-world conditions such as poor lighting, adverse weather, and dynamic environments often lead to corrupted camera images, posing significant challenges for reliable sensor fusion and downstream perception. In this paper, we propose Robu-MARC (Robust Masked Autoencoder-Aided Radar Camera), a fusion framework designed to enhance 3D perception in autonomous vehicles under sensor corruption. Robu-MARC integrates a Masked Autoencoder (MAE) with a Vision Transformer backbone to reconstruct degraded camera images and compute reconstruction error. This error serves a dual purpose; It weights the confidence attention map used in bird's eye view fusion and is incorporated into the loss function to guide training to corruption-tolerant spatial representations. On the radar side, Robu-MARC introduces a radar-specific cross-attention mechanism and applies Doppler-aware and radar cross-section (RCS)-aware Gaussian expansion strategies independently. By avoiding joint modeling of Doppler velocity and radar cross-section, the model improves target detection and enhances the reliability of multimodal fusion in real-world driving scenarios. We evaluate Robu-MARC on the nuScenes dataset and its corrupted variants, including scenarios with corrupted camera images. The performance of Robu-MARC is promising in object detection task across clean and corrupted images. This work advances robust multimodal fusion for autonomous driving and highlights the effectiveness of reconstruction-guided attention and selective radar feature refinement through Doppler- and RCS-aware processing in handling corrupt images.

## 1 INTRODUCTION

With an average of over 15 days per year spent behind the wheel and commuting times continuing to increase, the need for autonomous vehicles (AVs) has never been more evident (AAA Foundation for Traffic Safety, 2024). Road traffic injuries remain a critical global health concern; according to the *World Health Organization Global Status Report on Road Safety 2023* Organization (2023), an estimated 1.19 million people died from road traffic accidents in 2021. This underscores the urgent need for safer and smarter transportation solutions (Sun et al., 2024). Fully autonomous driving systems are seen as a promising remedy, mainly because they are immune to human limitations such as distraction, fatigue, and rule violations. This has spurred a growing interest in intelligent systems capable of anticipating and responding proactively to unseen or partially observed road participants.

Modern autonomous vehicle perception systems leverage a diverse suite of sensors—including cameras, LiDAR (light detection and ranging), radar (radio detection and ranging), and ultrasonic sensors—to construct a comprehensive understanding of the driving environment (Yeong et al., 2021). However, no single sensor guarantees reliable performance under all conditions. Cameras provide high-resolution data, but are vulnerable to adverse conditions such as glare, darkness, or fog. Radar is cost-effective and robust in bad weather, yet it suffers from low spatial resolution and data sparsity (Bilik et al., 2019). In addition, radar sensors are prone to misalignment issues that can further reduce perception accuracy, as highlighted in recent surveys (Sharif et al., 2025; Burza, 2024). To overcome these individual limitations, sensor fusion, particularly between cameras and radars, has become a key research focus. This approach takes advantage of the strengths of each modality to

improve the precision and robustness of detection, especially in challenging scenarios (Yao et al., 2024).

Despite these advancements, real-world deployments remain vulnerable to sensor corruptions such as low light, brightness changes, color quant, blurry, fog, and snow weather, as shown in Figure 1. RoboBEV benchmark Xie et al. (2023)demonstrated the significant degradation of BEV-based perception systems, including radar-camera fusion models, under such corruptions. The observed degradation under sensor corruptions highlights the need for models with stronger generalization capabilities as shown in Table 2

Building on this insight, we propose Robu-MARC, an enhanced version of the Camera-Radar Network (CRN) Kim et al. (2023) that integrates a Masked Auto-encoder (MAE) He et al. (2022) into its visual backbone. Our approach leverages a Bird's Eye View (BEV)-guided masking strategy during pretraining, which encourages the Vision Transformer to learn semantically rich and resilient spatial representations from corrupted inputs. This pretraining enables the model to effectively reconstruct missing or degraded visual information, thereby improving downstream fusion and 3D perception performance. We evaluated Robu-MARC on both clean and corrupted variants of the nuScenes dataset Caesar et al. (2020), focusing on challenging low-light and overexposure scenarios across varying difficulty levels. Our experiments show that Robu-MARC significantly recovers performance lost by baseline models in these degraded conditions, confirming the effectiveness of MAE-enhanced training for robust sensor fusion.

The main contributions of this work are summarized as follows:

- We propose **Robu-MARC**, a radar-camera fusion model that integrates BEV-guided Masked Auto-encoder (MAE) pretraining to improve robustness under sensor corruption.

- To capture fine-grained motion and reflectivity cues, we extract Doppler and radar cross-section (RCS) features from radar sensors and enrich them independently using a 3D Gaussian expansion strategy, enabling more expressive spatial-temporal representations.

- We perform comprehensive evaluations on clean and corrupted nuScenes datasets, demonstrating substantial improvements in NDS and mAP across different types of corruption.

- Scalability Across Corruption Severity Levels: We benchmark Robu-MARC on full-scale nuScenes datasets spanning multiple corruption severities—*easy*, *medium*, and *hard*—demonstrating its scalability and robustness in diverse real-world conditions.

Recent advances like Lift-Splat-Shoot Philion & Fidler (2020) and BEVFormer Li et al. (2022b) further demonstrate the effectiveness of BEV in multicamera and multisensor fusion. Therefore, our method leverages BEV-guided masking within the MAE pre-training process to improve spatial feature robustness, which is critical under corrupted sensor conditions. Our approach introduces a stronger inductive bias into the CRN framework, enhancing its robustness in scenarios involving missing, occluded, or degraded sensor data. We evaluate the proposed Robu-MARC on both clean and corrupted variants of the nuScenes dataset Caesar et al. (2020), with a particular focus on challenging low-light (Dark) and brightness corruption scenarios, as categorized in RoboBEV (Xie et al., 2023). For granular analysis, the test samples are divided into three difficulty levels: easy, medium, and hard. Baseline experiments reveal significant drops in detection performance (e.g., NDS from 0.3955 to 0.0511 under hard Dark scenarios). Our enhanced Robu-MARC model recovers much of this lost performance, confirming the benefits of MAE-enhanced training for sensor-fusion perception systems.

To provide a comprehensive understanding of our contributions, the remainder of this paper is organized as follows. We begin by reviewing recent advances across seven key areas, including transformer-based 3D object detection and perception, sensor fusion with BEV representations, camera-only and radar-only methods, camera-radar fusion strategies, masked autoencoders for multimodal learning, and the motivation behind our proposed Robu-MARC framework. Next, we present the proposed methodology, detailing the model architecture, the data preparation process, and the training strategies. We then describe the experimental setup, including the data sets used, the evaluation metrics, and the results under both clean and corrupted conditions. Finally, we conclude by summarizing key findings and outlining directions for future research.

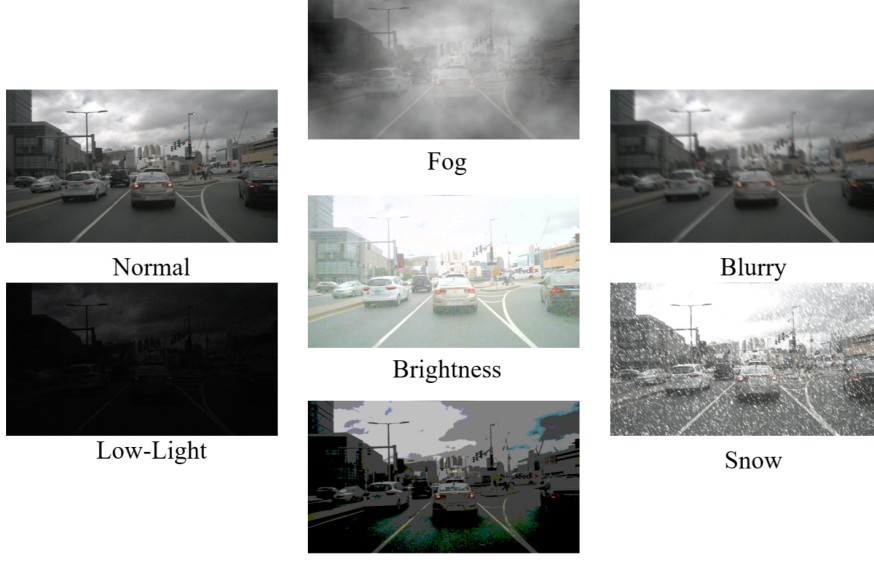

Figure 1: Sample front-facing camera images from nuScenes corrupt dataset illustrating variations across weather and lighting conditions.

## 2 RELATED WORK

### 2.1 TRANSFORMERS IN 3D OBJECT DETECTION AND PERCEPTION

Transformers have become foundational in deep learning for vision tasks due to their self-attention mechanism, which enables models to capture global context and long-range dependencies (Vaswani et al., 2017). Vision Transformers (ViTs) (Dosovitskiy et al., 2021) have proven particularly effective in 3D perception tasks, including object detection for autonomous driving, where multi-view and temporal cues are crucial. The use of transformers has improved perception robustness, particularly when processing complex multimodal sensor data.

Transformer-based architectures have recently shown great promise in this domain due to their ability to model long-range dependencies and effectively integrate heterogeneous data. Their core mechanism, *self-attention*, allows the network to dynamically weigh the importance of features across the entire input, enabling context-aware fusion across sensor modalities (Vaswani et al., 2017). Recent theoretical analysis has further unveiled the internal structure of self-attention via kernel principal component analysis (PCA), leading to the Robust Principal Component Attention (RPC-Attention) mechanism (Teo & Nguyen, 2024). By reformulating self-attention through kernel PCA, RPC-Attention emphasizes the principal components of the feature space while suppressing noisy or redundant signals, thereby improving robustness to input contamination. This provides valuable insights for designing transformer-based fusion models that must operate reliably under degraded sensor inputs. Although transformer-based architectures have shown great promise in multimodal sensor fusion due to their ability to model long-range dependencies and enable context-aware feature integration (Vaswani et al., 2017), their real-world deployment remains vulnerable to sensor corruptions. As demonstrated by RoboBEV Xie et al. (2023), BEV-based perception systems(Philion & Fidler, 2020), including fusion models such as, experience significant performance degradation under conditions such as low light, brightness shifts , and adverse weather.

### 2.2 SENSOR FUSION AND BEV REPRESENTATIONS

To overcome the limitations of individual sensors, recent works emphasize fusing modalities such as camera, radar, and LiDAR. Public datasets like nuScenes Caesar et al. (2020), Waymo Sun et al. (2020), and KITTI Geiger et al. (2013) have played a critical role in advancing fusion research by providing multimodal benchmarks. Recent advances also adopt Bird's-Eye View (BEV) represen-

tations Huang et al. (2023) to unify spatial features across sensors for improved localization and detection. Among these, camera-radar fusion has gained attention for leveraging camera semantics and radar's robustness under poor lighting and weather.

## 2.3 CAMERA-ONLY PERCEPTION METHODS

Camera-only 3D perception has advanced through BEV transformations and depth-aware modeling, often using transformers and self-supervised strategies. BEVFormer Li et al. (2022b) generates BEV representations from multi-camera images via spatiotemporal transformers. BEVDepth Li et al. (2022a) adds explicit depth supervision with a refinement module for monocular views, while BEVStereo Li et al. (2022c) incorporates stereo cues for dense depth maps. PETRv2 Liu et al. (2022) leverages multi-scale deformable attention for improved 3D query generation, and Fiery Hu et al. (2021) predicts future occupancy with temporal BEV segmentation from monocular inputs. While these methods show strong camera-based 3D perception, their robustness under corrupted sensor inputs remains to be systematically evaluated, highlighting the need for multimodal fusion strategies such as camera-radar fusion.

## 2.4 RADAR-ONLY PERCEPTION METHODS

Radar-based 3D detection benefits from robustness to weather and lighting but faces challenges from sparse and noisy returns. RadarDistill Yang et al. (2024) improves radar-only detection via LiDAR-guided feature and proposal distillation. LEROjD Palmer et al. (2024) leverages cross-modal training with LiDAR supervision to enhance radar perception, while RadarDETR Sun et al. (2023) applies transformer-based detection to sparse radar BEV maps. These methods demonstrate progress in radar perception, yet their robustness under corrupted radar signals or challenging real-world conditions remains to be evaluated, motivating multimodal fusion approaches for reliable autonomous perception.

## 2.5 CAMERA-RADAR FUSION METHODS

Camera–radar fusion leverages the high-resolution semantics of cameras with the robustness of radar under adverse weather and lighting. Recent works adopt transformer architectures and Bird's Eye View (BEV) representations to improve multimodal alignment. REDFormer Zhang et al. (2023) evaluates fusion in rain and night, highlighting radar's value when vision degrades, while CRN Kim et al. (2023) enhances BEV fusion through deformable attention and radar supervision.

RCBEVDet Lin et al. (2024a) introduces a radar-specific BEV encoder with cross-attention fusion, achieving strong results on nuScenes and VoD. Its extension, RCBEVDet++ Li et al. (2024), integrates sparse fusion and query-based modeling, improving both detection and BEV segmentation. RaCFormer Chu et al. (2025) further advances fusion with radar-guided depth, polar query initialization, and temporal encoding, delivering state-of-the-art detection under dynamic environments.

To address robustness, RobuRCDet list (2025) proposes a 3D Gaussian Expansion (3DGE) module and weather-adaptive fusion, showing resilience against noise and adverse conditions. Despite these advances, robustness under diverse corruptions remains underexplored, motivating evaluations such as those in Table 2.

## 2.6 MASKED AUTOENCODERS (MAE) FOR BEV AND SENSOR FUSION

Masked Autoencoders (MAEs) have emerged as a self-supervised paradigm for robust 3D perception, reducing dependence on labeled data and improving generalization across modalities. Recent works adopt MAE-style architectures for BEV representation learning: **BEV-MAE** Lin et al. (2024b) uses BEV-guided masking and point token reconstruction, explicitly evaluating robustness under corrupted inputs such as rain, fog, nighttime, and sensor dropouts. Other methods like M-BEV (Chen et al., 2024), LetsMap Gosala et al. (2024), and BEVPose Hosseinzadeh & Reid (2024) improve cross-modal learning or reduce labeled data requirements, but do not systematically test robustness under environmental or sensor corruptions.

**Robu-MARC** fills this gap by integrating a BEV-guided MAE into a camera-radar fusion pipeline, enabling robust multimodal feature extraction under adverse conditions, including rain, darkness, and sensor noise, and enhancing reliable perception for autonomous vehicles.

## 2.7 MOTIVATION FOR ROBU-MARC

Despite the growing adoption of transformer architectures and BEV representations in 3D perception, our review of related work reveals a critical limitation: most camera-only, radar-only, and camera-radar fusion models have not been explicitly evaluated under sensor corruption. This includes common real-world challenges such as low-light conditions, brightness shifts, occlusions, or weather-induced noise. As autonomous driving systems must function reliably in such degraded environments, robustness remains an underexplored yet essential aspect of perception modeling. Recent efforts, such as BEV-MAE Lin et al. (2024b), have demonstrated that Masked Autoencoders (MAE) can enhance the robustness in LiDAR-based BEV detection by self-supervised pre-training. However, MAE has not yet been integrated into radar-camera fusion frameworks, leaving a significant gap in robust multimodal perception research. To address this, we propose **Robu-MARC**, a corruption-aware radar-camera fusion framework that integrates BEV-guided Masked Autoencoder (MAE) pretraining into the Camera-Radar Network (CRN) (Kim et al., 2023). By aligning MAE's ability to learn semantically rich and generalizable visual representations with CRN's multimodal fusion pipeline, our approach enhances the visual backbone's capacity to extract context-aware and resilient features even when sensor inputs are partially missing or severely degraded. Through this design, Robu-MARC aims to bridge the gap between high-performance fusion models and real-world robustness, contributing a scalable and corruption-tolerant perception system for autonomous vehicles.

## 3 METHODOLOGY

### 3.1 PROPOSED MODEL ARCHITECTURE

We propose **Robu-MARC**, an enhanced transformer-based camera-radar fusion model specifically designed to improve 3D object detection performance in autonomous driving scenarios under both clean and corrupted sensor conditions. Building upon the foundation of the Camera-Radar Network (CRN) Kim et al. (2023), our model strategically integrates a Masked Autoencoder (MAE) pre-trained Vision Transformer He et al. (2022) as the primary visual backbone, thereby enhancing the model's ability to reconstruct and process degraded visual inputs.

The architecture comprises several key components that work synergistically to achieve robust multimodal perception:

Our architecture integrates several key components to ensure robustness under sensor degradations. At its core, a MAE-pretrained Vision Transformer enables effective camera feature extraction by reconstructing corrupted visual inputs through learned representations. A lightweight radar backbone complements this by efficiently capturing spatial, velocity, and angular features from radar point clouds while preserving computational efficiency. Both modalities are spatially aligned through a Bird's Eye View (BEV) projection, which provides a unified coordinate system for seamless cross-modal integration. To further enhance robustness, the Enhanced Cross-Attention Multi-layer Fusion (CAMF) module employs confidence-aware attention mechanisms that dynamically adapt fusion weights based on real-time sensor quality indicators such as brightness levels and signal-to-noise ratios. Finally, a multi-task detection head generates 3D bounding boxes, class confidence scores, and optional velocity predictions, while also supporting BEV semantic segmentation to enable consistent multi-task learning. Collectively, this architectural design allows the model to maintain high detection accuracy across diverse and challenging environmental conditions.

### 3.2 DATASET AND CORRUPTED DATA SOURCES

We conduct comprehensive evaluation of our approach using the nuScenes dataset Caesar et al. (2020), which provides rich multi-sensor data collected from six strategically positioned cameras and five radar sensors, accompanied by precise 3D bounding box annotations across 10 distinct object classes commonly encountered in urban driving scenarios. Performance assessment utilizes

standard autonomous driving metrics, including mean Average Precision (mAP) and the nuScenes Detection Score (NDS), which provide comprehensive evaluation of detection quality and localization accuracy.

To rigorously assess model robustness against real-world sensor degradations, we leverage corrupted camera data provided by the RoboBEV benchmark Xie et al. (2023), which systematically simulates adverse environmental conditions commonly encountered in autonomous driving. These include dark conditions, representing low-light and nighttime scenarios where camera sensors suffer from insufficient illumination, and bright conditions, which model overexposure and glare effects caused by direct sunlight or intense light sources

In our experimental framework, corrupted camera inputs undergo reconstruction using our **MAE-based visual backbone**, which leverages learned representations to recover meaningful visual information from degraded inputs. Similarly, corrupted radar inputs are processed through our proposed radar restoration methodology, ensuring both modalities receive appropriate preprocessing before fusion. The reconstructed and restored sensor modalities are subsequently integrated through our CRN-based late fusion framework, enabling comprehensive evaluation of our **Robu-MARC** architecture's robustness under both authentic camera degradations and systematically simulated radar corruptions.

## 3.3 FEATURE EXTRACTION AND PROJECTION

**Camera Encoder:** We adopt a ViT-B/16 Vision Transformer as the camera backbone, pretrained with our BEV-guided Masked Autoencoder (MAE). During pretraining, 75% of BEV-projected image patches are masked, forcing the model to capture long-range spatial dependencies and contextual priors. This enhances the encoder's ability to reconstruct corrupted camera inputs while preserving fine-grained spatial structure critical for 3D localization.

**Radar Encoder:** Radar returns - comprising range, azimuth, and Doppler - are first transformed into pseudoimage representations and encoded via a lightweight CNN backbone. The radar encoder produces feature maps in BEV space, spatially aligned with camera-derived features to ensure consistent fusion. This design maintains the quality of the feature under both clean and corrupted radar input, as illustrated in Figure 2. Following encoding, radar point features are aggregated within spatial bins and aligned to a BEV grid, enabling structured integration with camera features. This spatial alignment step ensures that radar data contributes meaningfully to downstream spatial reasoning in the transformer-based detection head.

**BEV Projection:** Camera and radar features are projected into a unified Bird's Eye View coordinate system using precise sensor calibration and geometric transformations. The resulting BEV features enable spatially aligned cross-modal interactions, providing a robust foundation for multimodal fusion.

## 3.4 SENSOR FUSION

We employ an enhanced Cross-Attention Multi-layer Fusion (CAMF) module, augmented with confidence-aware attention that dynamically weights features based on modality reliability. Quality metrics, including visual clarity, radar SNR, and cross-modal consistency guide the attention mechanism, allowing the model to prioritize reliable signals while mitigating degraded inputs. This adaptive fusion ensures robustness under sensor corruption and supports effective integration of MAE-reconstructed camera features and radar data.

## 3.5 OBJECT DETECTION HEAD

Fused BEV features are decoded via a transformer-based detection head, producing 3D bounding boxes, class probabilities, and optional velocity estimates. Multi-task learning is incorporated via BEV semantic segmentation, promoting spatial consistency and improving generalization. The transformer decoder leverages cross-attention between BEV queries and fused features, enabling precise object localization under challenging sensor conditions, as illustrated in the architecture of Robu-MARC (see Fig. 2)

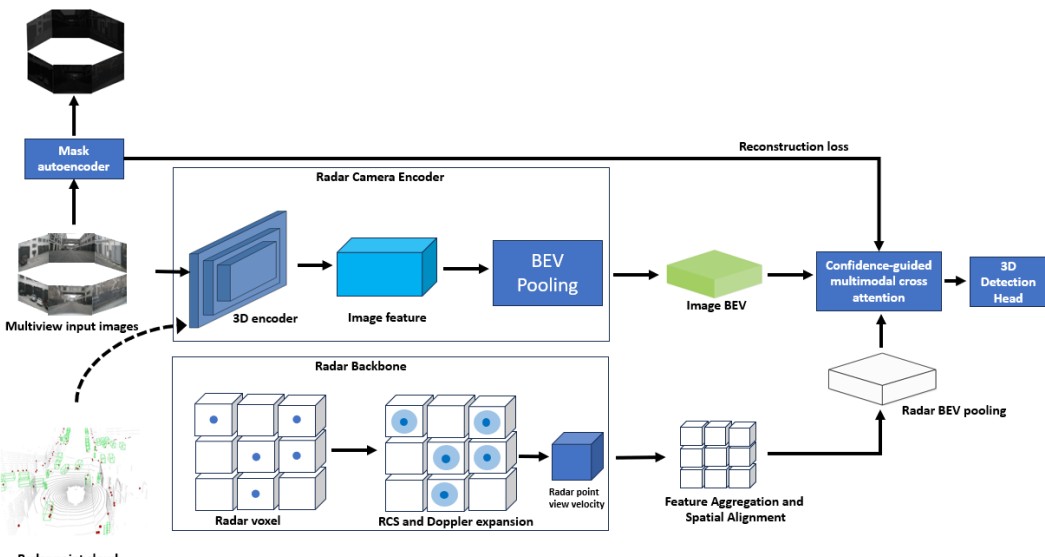

Figure 2: Architecture of the Masked Autoencoder-Aided Camera Radar Net (Robu-MARC) for robust 3D object detection under sensor corruption. The radar point cloud visualization is adapted from Nabati & Qi (2020).

## 3.6 Gaussian Expansion, RCS Aware BEV

This section details the proposed 3D Gaussian Expansion (3DGE) module. Building upon the 3DGE approach introduced in RobuRCDET, we propose an improved module that employs an expanding Gaussian kernel to better densify radar features, leverages a deformable kernel map, and restricts kernel size to balance efficiency and accuracy. First, we input the RCS and velocity information for each radar point into a parameter encoder, which generates a deformable kernel map and determines the variance of the Gaussian kernel. Next, we apply 3D Gaussian expansion to each radar point. Specifically, the RCS and velocity values are expanded into the surrounding voxels of each radar point, with the spreading range determined by the kernel size $\lambda_p$ provided by the deformable kernel map. To balance efficiency and accuracy, we restrict that $\lambda_p \in \{1, 3, 5\}$. After expansion, the RCS and velocity values are summed within each voxel to create a complete feature volume.

This process is formally described by the following equations:

$$V_v^{3DGE}(x, y, z, v) = \frac{V(vE)}{2\pi\sigma_v^2} \left\{ \exp\left( \frac{|x - x_v|^2 + |y - y_v|^2}{2\sigma_v^2} \right) \cdot |x - x_v| \in \Delta_v, |y - y_v| \notin \Delta_v \right\} \tag{1}$$

$$V_{res}^{3DGE}(x, y, z, RCS) = \frac{V(RCS)}{2\pi\sigma_{res}^2} \left\{ \exp\left( \frac{|x - x_{res}|^2 + |y - y_{res}|^2}{2\sigma_{res}^2} \right) \right.$$
$$\left. \cdot [|x - x_{res}| \in \Delta_{res}, |y - y_{res}| \in \Delta_{res}] \right\} \tag{2}$$

$$V(x, y, z, RCS, v) = V_{res}^{3DGE}(x, y, z, RCS) + V_v^{3DGE}(x, y, z, v) \tag{3}$$

Where $V$ represents the radar voxel, $x_p$ and $y_p$ are the $x$ and $y$ coordinates of the radar point, and we perform the expansion on the RCS and velocity dimensions. Finally, the total 3DGE result, $V^{3DGE}(x, y, z, RCS, v)$, combines the three expanded volumes as a res-block manner for downstream processing.

### 3.7 LOSS FUNCTIONS

Our training optimization employs a composite loss function integrating multiple objectives to enhance both detection accuracy and robustness. L1 Loss is used for 3D bounding box regression, quantifying differences between predicted and ground truth parameters including center coordinates, dimensions, and orientation angles. Intersection over Union (IoU) Loss evaluates spatial overlap quality between predicted and ground truth 3D boxes, with variants such as GIoU, DIoU, or CIoU to improve alignment. Cross-Entropy Loss is applied to Bird's Eye View (BEV) semantic segmentation, measuring divergence between predicted class probabilities and ground truth labels. An optional Velocity Loss minimizes errors for object motion predictions, supporting accurate tracking of dynamic objects.

The overall training loss is expressed as a weighted combination:

$$\mathcal{L}_{\text{total}} = \lambda_1 \mathcal{L}_{\text{L1}} + \lambda_2 \mathcal{L}_{\text{IoU}} + \lambda_3 \mathcal{L}_{\text{CE}} + \lambda_4 \mathcal{L}_{\text{Velocity}}, \tag{4}$$

where $\lambda_1$, $\lambda_2$, $\lambda_3$, and $\lambda_4$ are hyperparameters tuned to balance the contribution of each loss component.

### 3.8 TRAINING AND EXPERIMENTAL SETUP

We implemented our Robu-MARC framework in PyTorch. The camera encoder is initialized with MAE-pretrained weights, while the radar backbone employs a compact 3-layer CNN. Training is conducted using the AdamW optimizer with a learning rate of $2 \times 10^{-4}$, weight decay of $1 \times 10^{-2}$, and batch size 4. A 14-epoch cosine annealing learning rate schedule is applied to stabilize convergence. Camera images are resized to $224 \times 224$ for MAE pretraining and $256 \times 704$ for CRN input. Data augmentation techniques, including random horizontal flipping, cropping, and brightness scaling, are used to improve generalization and robustness.

All experiments are performed on a single NVIDIA RTX A6000 GPU, which provides sufficient computational resources for both training and inference while ensuring reproducibility across research environments. To enrich radar input while maintaining architectural simplicity, we incorporate five radar sweeps: one temporally aligned with the current camera frame and four preceding sweeps that provide historical context. This temporal accumulation strategy significantly enhances radar point density without introducing complex recurrent or attention-based temporal fusion mechanisms. Since radar sensors typically operate at higher sampling frequencies than cameras, the accumulated sweeps often include measurements captured after the previous camera frame, ensuring temporal relevance and coherence with the current visual observations.

Our Robu-MARC design deliberately avoids heavy temporal fusion beyond radar sweep accumulation. This allows us to isolate the specific contribution of BEV-guided MAE pretraining in improving robustness and detection performance, without confounding effects from additional temporal modeling.

Table 1: 3D detection results comparison on the nuScenes validation set using ResNet50 backbone. 'C' , 'R' represent camera and radar respectively. Epochs indicate training iterations.

| Method | Modality | Backbone | Image Size | Epochs | NDS↑ | mAP↑ | mATE↓ | mASE↓ | mAOE↓ | mAVE↓ | mAAE↓ |
|---|---|---|---|---|---|---|---|---|---|---|---|
| CRN Kim et al. (2023) | C+R | ResNet50 | 256×704 | 24 | 56.0 | 49.0 | 0.487 | 0.277 | 0.542 | 0.344 | 0.197 |
| RobuRCDet list (2025) | C+R | ResNet50 | 256×704 | 24 | 56.7 | 51.2 | 0.481 | 0.273 | 0.499 | 0.317 | 0.193 |
| RaCFormer Chu et al. (2025) | C+R | ResNet50 | 256×704 | 24 | 61.3 | 54.1 | 0.478 | 0.261 | 0.449 | 0.208 | 0.180 |
| RCBEVDet Lin et al. (2024a) | C+R | ResNet50 | 256×704 | 24 | 56.8 | 45.3 | 0.486 | 0.285 | 0.404 | 0.220 | 0.192 |
| RCBEVDet++ Li et al. (2024) | C+R | ResNet50 | 256×704 | 24 | 60.4 | 51.9 | 0.488 | 0.268 | 0.408 | 0.221 | 0.177 |
| RobuRCDet list (2025) | C+R | ResNet50 | 256×704 | 14 | 50.9 | 42.2 | 0.559 | 0.289 | 0.631 | 0.350 | 0.180 |
| Robu-MARC (Ours) | C+R | ResNet50 | 256×704 | 14 | **46.5** | **40.6** | **0.576** | **0.296** | **0.645** | **0.626** | **0.238** |

### 3.9 EVALUATION METRICS

To evaluate the performance of our 3D object detection model, we adopt the official metrics defined by the nuScenes benchmark (Caesar et al., 2020), which assess both classification accuracy and geometric precision under diverse environmental conditions. Mean Average Precision (mAP) measures

Table 2: Performance comparison of RobuRCDet and Robu-MARC (ours), both trained for 14 epochs, under different corruption conditions and difficulty levels on the nuScenes validation set.

| Corruption | Difficulty | Method | Epochs | NDS↑ | mAP↑ | mATE↓ | mASE↓ | mAOE↓ | mAVE↓ | mAAE↓ |
|---|---|---|---|---|---|---|---|---|---|---|
| Low Light(Dark) | Easy | RobuRCDet | 14 | 32.0 | 17.6 | 0.705 | 0.311 | 0.835 | 0.615 | 0.208 |
| | | Robu-MARC (Ours) | 14 | **29.2** | **16.9** | **0.747** | **0.314** | **0.813** | **0.768** | **0.276** |
| | Mid | RobuRCDet | 14 | 27.2 | 12.5 | 0.739 | 0.323 | 0.894 | 0.736 | 0.207 |
| | | Robu-MARC (Ours) | 14 | **25.1** | **11.7** | **0.786** | **0.323** | **0.838** | **0.844** | **0.283** |
| | Hard | RobuRCDet | 14 | 20.4 | 6.53 | 0.785 | 0.332 | 0.955 | 1.019 | 0.204 |
| | | Robu-MARC (Ours) | 14 | **19.5** | **6.10** | **0.840** | **0.341** | **0.866** | **1.012** | **0.305** |
| Brightness | Easy | RobuRCDet | 14 | 48.2 | 38.1 | 0.579 | 0.302 | 0.651 | 0.367 | 0.183 |
| | | Robu-MARC (Ours) | 14 | **42.1** | **35.9** | **0.608** | **0.304** | **0.608** | **0.726** | **0.262** |
| | Mid | RobuRCDet | 14 | 44.1 | 31.8 | 0.604 | 0.315 | 0.673 | 0.405 | 0.181 |
| | | Robu-MARC (Ours) | 14 | **38.6** | **29.8** | **0.619** | **0.308** | **0.704** | **0.753** | **0.244** |
| | Hard | RobuRCDet | 14 | 40.7 | 26.7 | 0.622 | 0.321 | 0.682 | 0.450 | 0.184 |
| | | Robu-MARC (Ours) | 14 | **34.2** | **24.5** | **0.641** | **0.312** | **0.756** | **0.832** | **0.259** |

the model's ability to detect and classify objects across all categories, computed using a center distance threshold (typically 0.5 meters) and averaged over all classes. The nuScenes Detection Score (NDS) is a composite metric that combines mAP with five true positive (TP) error metrics: translation, scale, orientation, velocity, and attribute errors, offering a holistic evaluation of the 3D object detection performance.

$$\text{NDS} = \frac{1}{10}\left[5 \times \text{mAP} + \sum_m \left(1 - \min(1, m)\right)\right]$$

where the summation is taken over the following TP error metrics: mATE (Mean Average Translation Error), mASE (Mean Average Scale Error), mAOE (Mean Average Orientation Error), mAVE (Mean Average Velocity Error), and mAAE (Mean Average Attribute Error).

### 3.10 DISCUSSION OF RESULTS

The experimental results indicate that Robu-MARC is a promising candidate for robust 3D object detection across both standard and corrupted conditions. As shown in Table 1, Robu-MARC achieves competitive performance with fewer training epochs (14), suggesting efficient learning and effective fusion of camera and radar modalities. Under corrupted scenarios in Table 2, Robu-MARC consistently performs well across multiple metrics, with notable improvements in mAP and velocity-related errors (mAVE), particularly under low-light and brightness-shift conditions. While certain error metrics remain elevated in more challenging settings, the model's stability and generalization capacity point to its potential for further gains with extended training and fine-tuning. These findings highlight Robu-MARC's resilience to sensor degradations and its suitability for real-world autonomous driving environments.

## 4 CONCLUSION

We presented Robu-MARC, a robust radar–camera fusion framework that combines a Masked Autoencoder with Doppler and RCS-aware radar feature refinement. By leveraging reconstruction error during BEV fusion and training, Robu-MARC learns corruption-tolerant spatial representations that improve multimodal 3D perception. Experiments on nuScenes and its corrupted variants show consistent gains under challenging conditions, validating the effectiveness of reconstruction-guided attention and selective radar processing.

Future work will address radar-specific noise such as ghost reflections, multi-path interference, and signal sparsity. We also aim to evaluate Robu-MARC across broader real-world scenarios and sensor degradation patterns to further assess its robustness in safety-critical environments.

## 5 ACKNOWLEDGEMENT

I would like to acknowledge the use of Microsoft Copilot in the preparation of this manuscript. Copilot provided valuable assistance in polishing the writing.

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
