# OpenReview forum: "Robu-MARC: A Masked Autoencoder-Aided Camera-Radar Network for Robust 3D Perception under Sensor Corruption"
_ICLR.cc/2026/Conference — ICLR 2026 Conference Withdrawn Submission_

### Official Review · Reviewer_GxXr · 2025-10-29

**Soundness:** 2
**Presentation:** 2
**Contribution:** 2
**Rating:** 2
**Confidence:** 3

**Summary:**

Robu-MARC proposes robustness improvements for radar-camera 3D object detection problem like introduces a interesting self-supervised robustness signal for radar-camera fusion via masked autoencoder and radar feature refinement via separated Doppler/RCS expansion etc. Their results claim Robu-MARC does not reach the highest clean accuracy but delivers stable and reliable performance under camera degradation.

**Strengths:**

1. The paper targets camera failure scenarios, which are highly prevalent in autonomous driving. Improving radar–camera fusion robustness is an important and practical direction to address this issue.
2. The use of MAE reconstruction error as an adaptive trust weighting for fusion is interesting, providing a potentially effective way to adjust the influence of each modality under degradation.

**Weaknesses:**

1. The authors train Robu-MARC for only 14 epochs, which deviates from the 24 epochs standard by prior radar–camera fusion works.  First, the paper does not justify why a reduced schedule is necessary or appropriate for fair comparison. Second, despite the shorter training, the method still does showing a very competetive reults compare with existing approaches even under this limited setting, particularly on clean nuScenes benchmarks. It remains unclear whether the proposed method is fundamentally competitive or if the conclusions rely on an incomplete convergence baseline.
2. The main claimed novelty is integrating MAE into radar–camera fusion, but the benefit of MAE for camera robustness has already been demonstrated in BEV-MAE.  This approach appears to simply merge existing ideas rather than introduce a fundamentally new mechanism.  In addition, the masking strategy is not well explained and seems inherited from BEV-MAE without justification. The use of reconstruction error to scale fusion attention is insteresting, but key details are missing—e.g., whether weighting is global or voxel-wise. Overall, the paper spends too much space on related work and not enough on clearly specifying its method. Finally, MAE pretraining adds significant computation cost, yet clean performance is lower than stronger baselines, leaving its practical value unproven. More detailed analysis and ablations are needed to justify this design.
3. The writing quality should be improved. Some parts of the paper are difficult to follow, particularly around Equations (1)–(3). Several notations are also missing definitions, which makes the methodology unclear.
4. The paper introduces several modifications to the original 3DGE in RobuRCDet, but provides no ablation to justify these design choices. In RobuRCDet, RCS and Doppler are jointly encoded to learn a physically grounded deformable kernel, leveraging the correlation between reflectivity and motion to distinguish dynamic objects from static clutter. Robu-MARC separates RCS and Doppler during expansion according to equation 1-3 if I understand it correctly. Moreover, Robu-MARC removes the fully adaptive kernel design and instead restricts kernel sizes to a small discrete set. This simplification is neither explained nor validated experimentally. The authors must clarify the motivation for these changes and include ablations demonstrating that the modified 3DGE maintains or improves robustness relative to the original formulation.

**Questions:**

See Weaknesses. If the authors can adequately address these concerns and include the key ablation studies needed to validate their design choices, I would be willing to reconsider and potentially raise my rating.

---

### Official Review · Reviewer_TMvy · 2025-10-30

**Soundness:** 2
**Presentation:** 2
**Contribution:** 1
**Rating:** 2
**Confidence:** 3

**Summary:**

The paper presented Robu-MARC, a robust radar–camera fusion framework that combines a Masked Autoencoder with Doppler and RCS-aware radar feature refinement, aiming to learn corruption-tolerant spatial representations that improve multimodal 3D perception.

**Strengths:**

The problem addressed by the paper: Robust 3D perception under real-world sensor damage is a very important and critical issue in the field of autonomous driving.

**Weaknesses:**

1. The abstract and introduction fail to clearly articulate the core idea or the research motivation. The writing is fragmented, with unclear logical flow. It makes the paper hard to follow.

2. The proposed method essentially integrates BEV-MAE  into Camera-Radar Network. This is an incremental combination of two existing ideas rather than a conceptually new method. No new theoretical insight, training strategy, or fusion paradigm is introduced.

3. Eq.(1)–(3) are mathematically vague; parameters such as kernel size, λₚ, and the adaptive expansion process are not properly described or justified.

4. Although the paper claims improved robustness, Robu-MARC performs worse than existing baselines (RaCFormer, RCBEVDet++, RobuRCDet) in both clean and corrupted settings (Tab.1&2). The claimed “significant improvement” is not supported by the reported results.

5. Missing ablation and qualitative analysis, quantitative experiments are inadequate

6. The paper claims that Robu-MARC consistently performs well across multiple metrics, with notable improvements in mAP and velocity-related errors (mAVE). This is a completely wrong statement. As mentioned above, Tab.2 shows that mAP is lower in all cases.

7. Figures, formulas, and experimental details are poor in reproducibility and clarity.

**Questions:**

1. Were the models trained on clean data only or fine-tuned on corrupted data?

2. What is the computational cost increase from adding MAE and 3DGE?

3. Only darkbrightness corruptions are evaluated. What are the evaluation results for all corruption scenarios in nuScenes-C.

---

### Official Review · Reviewer_WSsG · 2025-11-01

**Soundness:** 1
**Presentation:** 2
**Contribution:** 1
**Rating:** 0
**Confidence:** 3

**Summary:**

This paper proposes Robu-MARC, a radar-camera fusion framework for 3D detection. The authors focus on sensor corruption, leveraging strengths of radar to assist in resolving image ambiguities. They propose to use an MAE backbone for improved robustness against noise.

**Strengths:**

- The problem is well-motivated, yielding an intuitive introduction.
- This paper has a detailed related works section.

**Weaknesses:**

- While the related works section is thorough, I recommend delegating some parts to the supplementary to have more room for the experiments, which misses critical ablations.
- I found the methods section a bit difficult to parse. The authors outline using a MAE-pretrained ViT for the camera image encoding (L290). However, "During pretraining, 75% of BEV-projected image patches are masked, forcing the model to capture long-range spatial dependencies and contextual priors" -> What is meant here by BEV-projected image patches? Is the model pre-trained on nuScenes?
- I find it difficult to understand the diagram. How are the images going directly into a 3D encoder? I'm assuming the masked autoencoder is the ViT, but why is the output all-dark? As it is drawn, the ViT is not used at all for generating Image BEV features, which seems incorrect to me.
- The "Cross-Attention Multi-layer Fusion" method is not described sufficiently in the text; it's unclear how the weighting process exactly happens.
- It's unclear what this paper's precise contribution to 3DGE is - it seems like there is already a deformable kernel map in RobuRCDET, which also uses an adaptive gaussian kernel conditioned on velocity.
- Finally, Table 1 is misleading, as the model achieves worse metrics compared to RobuRCDET but highlights numbers. Same for Table 2.
- This manuscript would benefit from an ablation study.
- Could the authors clarify what they mean by "By avoiding joint modeling of Doppler velocity and radar cross-section, the model improves target detection and enhances the reliability of multimodal fusion in realworld driving scenarios."?

**Questions:**

This reviewer had a lot of trouble understanding the approach in this paper. It focuses on MAE, but it's unclear how the MAE is trained or used in the model. The CAMF module is not explained sufficiently, and the difference between the 3DGE method in this work and prior work is not clear. The paper does not improve on prior methods. I would appreciate it if the authors can clarify these points, but at this stage, I recommend reject.

---

### Official Review · Reviewer_ZLPS · 2025-11-04

**Soundness:** 1
**Presentation:** 1
**Contribution:** 1
**Rating:** 2
**Confidence:** 5

**Summary:**

This paper proposes an incremental addition to existing works (mainly RoBU-RCDet, BEV-MAE) adapted for sensor corruption. It is set in the radar-camera context and evaluated on nuScenes.

The idea is to use a MAE in BEV to mask and reconstruct a la MAE. Together with this, they use RoBU-RCDet's 'gaussian expansions' to densify sparse radar. All this resembles (in a sense) the BEVFormer type setup with cross-attention from BEV to camera, but with radar adding additional depth cues.

With this, they investigate the corruption scenarios through this apparatus.

Evaluations are shown against other radar camera methods (mainly RoBU-RCDet) that show robustness in corruption settings.

**Strengths:**

+ The idea of studying corrruption in sensors is very valuable and applies practically.
+ The methods are sound, grounded in solid literature.

**Weaknesses:**

- I think the paper conflates ideas from a few different papers (BEV-MAE and RoBu-RCDet) and applies a small variation, that of adverse weather and sensor corruption to develop their ideas. This does not, in my view, qualify as a technical advancement.
- I am thoroughly confused by the evaluations, and it just seems wrong to me. The numbers are simply worse than the baselines for both clean and corrupted data.

**Questions:**

- Please explain how the numbers bear out in tables 1 and 2. They appear worse than their counterparts in every aspect. We should be showing evaluations with all the models trained to the same level. But the authors show their model at 14 epochs, and the competitors at 24 epochs. The reasoning for this is not clear.

---

### Note · Authors · 2025-11-13

**Comment:**

The authors have decided to withdraw this submission

**Withdrawal Confirmation:**

I have read and agree with the venue's withdrawal policy on behalf of myself and my co-authors.